# Characteristics of Standing Postural Control in Women under Additional Load

**DOI:** 10.3390/ijerph17020490

**Published:** 2020-01-13

**Authors:** Bożena Wojciechowska-Maszkowska, Dorota Borzucka

**Affiliations:** Faculty of Physical Education and Physiotherapy, Opole University of Technology, 45-758 Opole, Poland; d.borzucka@po.edu.pl

**Keywords:** stability control, posturography, external load, women

## Abstract

The aim of this study was to evaluate the effect of additional load on postural-stability control in young women. To evaluate postural control in the 34 women in this study (mean age, 20.8 years), we measured postural sway (center of pressure, COP) in a neutral stance (with eyes open) in three trials of 30 s each. Three load conditions were used in the study: 0, 14, and 30 kg. In analysis, we used three COP parameters, variability (linear), mean sway velocity (linear), and entropy (nonlinear). Results suggested that a considerable load on a young woman’s body (approximately 48% of body weight) had significant influence on stability. Specifically, heavy loads triggered random movements, increased the dynamics of postural-stability control, and required more attention to control standing posture. The results of our study indicate that inferior postural control mainly results from insufficient experience in lifting such a load.

## 1. Introduction

The main objectives of posturographic research are to understand and define the impact of various factors that distort the stability of postural changes and contribute to the risk of collapse [1]. Little research has been done on the impact of a long-term mechanical load on motor activity and postural stability in the human body. These studies focused on the effects of a load on static and dynamic balance control [2,3,4,5,6]. Recent studies, such as those by Crommert et al. [7] and Costello et al. [8], showed that a load of certain value affects the stability of the human body. Optimal postural control requires sensory information from the visual, vestibular, and proprioceptive systems. Studies on the effect of static loads on standing posture showed intense activation of the proprioceptive system. This discovery is explained in different ways with regard to balance-control disorders [9]. Other researchers have assessed the effect of external loads on postural-stability control in various load configurations (carrying a backpack or placing extra load around the waist) [2,3].

There is little scientific research in the correlation between a woman’s profession and load impact on her postural control. Nowadays, women do harder physical work more often than in the past for roles formerly considered to be male occupations. In fields such as physiotherapy, medical rescue, law enforcement, the military, and firefighting, portable rescue equipment weighs in the range of 10–15 kg. Therefore, further research in this area is justified.

### 1.1. Study Objective

The aim of this study was to evaluate the effect of symmetric loading of different intensities on balance control in young women. Analysis of the impact of various loads on body movements in young women can provide a better understanding and explanation of postural control and can improve the handling of physical loads in women in the workplace. A strong evidence base is important regarding the physical activity of women working in jobs that involve external loads. Handling a large symmetric load is an atypical and difficult task for the nervous system, and could have a negative impact on balance.

### 1.2. Research Question

This study aimed to answer one research question: what impact does an additional external load have on the balance control in young women?

### 1.3. Hypotheses

The following hypotheses are proposed:the dynamics of body control change under the influence of an added external load;greater postural load on young women results in significant changes to balance control; anda large external load causes a deterioration in standing postural control (increase in center of pressure (COP) parameters) and a decrease in entropy (associated with the difficulty of the task).

## 2. Materials and Methods

### 2.1. Participants

Before the test, participants were familiarized with the aim, methodology, and procedures of the study. They signed a written informed consent form in order to participate in the study. Approval from the local Bioethics Committee was obtained, and all experiments were performed according to the Helsinki Declaration.

In accordance with the goals of the study, the participants were young women. All participants in the study had a positive medical certificate for their psychophysical fitness, which enabled them to enter the study. An occupational physician assessed participants’ ability to perform submaximal and maximal efforts under different environmental conditions. The basic medical examination included a neurological examination and tests of vision, hearing, the heart, and morphology. Specifically, 34 adult women (physiotherapy students) with an average age of 20.8 ± 1.8 years, body height of 167.7 ± 5.1 cm, and body weight of 62.0 ± 7.9 kg participated in the study. The exclusion criteria for participants in the study were as follows: no injuries, dizziness, or disease. Participants undertook various forms of recreational physical activities (1 h twice per week as part of the teaching program at the university).

### 2.2. Apparatus

Postural control was assessed on a force platform (Type 9286AA, Kistler Instrumente AG, Winterthur, Switzerland) with a sampling frequency of 100 Hz.

### 2.3. Research Procedures

The study consisted of 3 conditions—L0 (no load), L1 (14 kg load with 7 kg in each hand—22.6% of mean body weight), and L2 (30 kg load with 15 kg in each hand—48.4% of mean body weight). Subjects held the load in their hands with a gripper because that was usually how the load was transferred to them. Subjects had their eyes open in all experiments.

The load (weights) was symmetrically distributed at the level of the trochanters of the larger thigh bones (femurs). Participants stood barefoot with a 14° angle between their feet and a distance of 17 cm between their heels [10]. The upper extremities were positioned along the side of the body (with their arms by their sides). Participants were instructed to stand as motionless as possible for 30 s, with their gaze focused on a brightly colored marker at eye level at a distance of 1.5 m [3].

### 2.4. Data Analysis

In analysis, we used 2 linear COP parameters (variability and mean sway velocity) and 1 nonlinear parameter (entropy). The temporary position of the COP of the feet was calculated from ground-reaction forces, and was analyzed in the medial–lateral (M–L) and anterior–posterior (A–P) directions. Linear and nonlinear parameters describing the control of postural balance in young women were calculated on the basis of the COP signal. Linear parameters included the standard deviation (SD) of the time series and the mean velocity (MV). Lower values for these parameters indicated a more efficient postural balance control. Sample entropy (SE), a nonlinear parameter, is a measure of the irregularity or unpredictability of a time series. It is associated with the amount of attention paid to a postural task and with the automatism related to performing that task [11,12].

### 2.5. Statistical Methods

COP signal parameters were explored using a Shapiro–Wilk test for the normality of the distribution. Since the distribution was similar to normal distribution, repeated-measures ANOVA was used to compare COP parameters across different tasks. The repeated-measure factors included 3 load levels (0, 14, and 30 kg) and 2 COP directions (anterior–posterior and medial–lateral). To compare differences between individual factors, we used a post hoc Tukey multiple-comparisons procedure (HSD test) with a significance level of *p* = 0.05.

## 3. Results

Table 1 shows descriptive statistics (M ± SD) for the COP for the three experiment conditions.

Table 2 shows analysis of variance using repeated-measures ANOVA. Figure 1 shows the main effects of the post hoc test.

Repeated-measures ANOVA revealed that the following parameters significantly affected postural stability: SD, MV, and SE.

The impact of the direction of movement on postural stability was found to be insignificant only for the SE parameter. A very high level of statistical significance was observed for SD and MV.

Results revealed that the load and direction of movement had no impact on postural-stability parameters (SD, MV, and SE).

Results of the post hoc Tukey (HSD) test revealed differences in postural-stability control between experiment conditions in both the A–P and M–L directions. Figure 1 shows differences between experiments with respect to SD, MV, and SE. Results revealed that the weight of the external load significantly affected postural control and the characteristics of the subjects’ sway. Postural control among participants decreased significantly in Experiment L2. The post hoc test showed that the highest load in Experiment L2 produced a significant increase in the variability of sway (SD) compared to the conditions in Experiments L0 and L1 (*p* ≤ 0.001 and *p* ≤ 0.002, respectively). Additionally, sway MV was significantly higher in Experiment L2 than that in L1 (*p* ≤ 0.001) and L0 (*p* ≤ 0.01). In contrast, entropy was significantly lower in Experiment L2 than that in Experiments L0 and L1 (*p* ≤ 0.001 and *p* ≤ 0.05, respectively).

## 4. Discussion

The aim of this study was to assess the impact of a load on postural-stability control in young women. The methodology used in this study and the studied COP parameters are well-documented in the specialist literature [13,14,15]. Research indicates that the neuromuscular system is more involved in the control of standing postural stability in the A–P direction than in the M–L direction [16,17]. Increased body fluctuation is one of the symptoms of posture-control deficits. Research conducted by Orawiec et al. [18] showed that wives of patients with Parkinson’s disease had greater body oscillations compared to women with different lifestyles (mean velocity and range of oscillations in both the sagittal and frontal directions). Additional load corresponds with physical activity in the different decades of human life. This load may be of a different nature (static and dynamic actions) [8,9,19,20]. This form of physical activity is associated with work (trade, medical work, emergency services, uniformed services, industry, construction, and agriculture), intensive sports, recreational activities, and typical daily work (shopping, carrying luggage, carrying various objects, etc.).

Other researchers concluded that instability increases linearly with increased external loading (8%, 22%, and 54% of body weight) [21]. The results of this study showed that young women had better control of their standing position under moderate loads than under higher loads. In this study, a small load (L1) did not significantly affect body sway (SD). The observed increase in body sway variation may have resulted from functional adaptations to elevated sensory thresholds in peripheral receptors and enhanced perception of sensory information integration [13,15]. According to Wegen et al. [21], the pattern of swaying, and the fact that balance control is not an isolated motor task, should be taken into account when evaluating the variability of swaying. Detailed analysis by Werner et al. [22] explained changes in biomechanical conditions during experiments with additional load. Results obtained by the aforementioned authors indicated that load and control conditions (closed eyes) cause a significant increase in instability. Other researchers concluded that instability increases linearly with increased external loading (8%, 22%, and 54% of body weight). In the A–P and M–L directions, the area of COP sway increases linearly with the weight of the load. This means that more control is needed to maintain postural stability [2]. The studies of Zultovsky and Ariun [3] showed that weight value (10% and 20% of body weight) and position, and the size of the support surface determine balance to a large extent. The results of these studies suggest that a symmetrically distributed external load (a backpack) is the most beneficial form of loading. This type of load does not destabilize standing posture. Conversely, an asymmetrical load causes rocking and increases the risk of falling. Bampouras and Dewhurst [6] did not observe a significant effect of loads on postural instability. However, maximal load weight was 9% of average body weight for older women and 8.5% for younger women (such loads correspond to motor activities similar to shopping).

Our data also indicated a change in kinematic stability control under the influence of a load. In standalone conditions (L0) and with little external load (L1), there was no increase in swaying in young women. However, higher loads (L2) contributed to a significant increase in the MV of swaying compared to L0 in the A–P (14.8%) and M–L (16.5%) directions, and to L1 in the A–P (20.2%) and M–L (21.09%) directions. The observed changes in sway characteristics could be explained by the need to increase muscle activity to stabilize the ankle joint [23,24,25] and by the effects of muscle fatigue resulting from the application of additional static load.

Results of other studies also confirmed that the increase in load contributes to the increase in COP speed [3]. In addition, these studies observed a large increase in COP velocity in the M–L direction in participants in a steel test with asymmetrical loads under limited support-base conditions (alloys held together). These discoveries may be important in developing ways of transferring loads in order to apply lighter loads to the human body and optimize postural control. Results of research conducted by Roster et al. [4] showed that changing the center of mass (COM) position changes COP amplitude parameters (velocity), especially in the A–P direction. The authors suggested that a change in COM position, obtained by modifying one or both physical parameters (increasing the difficulty of functional tasks), may be a promising way of developing new methods of training aimed at balance and functionality. Moreover, earlier studies confirmed that load weight affects postural control and crocus antigravity muscle activity [20,25]. An increase in body sway was observed with loads that induced a muscle-contraction force of 45% of the maximum. Other studies confirmed that a backpack load applied to the body affects dynamic balance and movement speed—movement speed decreased in all directions when carrying the backpack. Moreover, control was better in the A–P direction than in the M–L direction [19].

Our results do not confirm the observations of Duarte et al. [26], who did not find any differences in COP patterns. In their studies, scientists applied an evenly distributed load of approximately 43% of body weight around the torso at hip level (closer to COM). Another study by Haddad et al. [27] did not confirm the effect of symmetrical loads and changes in asymmetrical COP parameters (light loads of 1–9 kg).

In our study, young women had to invest more attention into controlling their standing position when their bodies were subjected to a significant external load (L2). This was reflected in a reduction in entropy. However, when no load was present (L0) and when a small load (L1) was applied, participants had greater control over their balance because these loads were similar to those of normal activities [12]. Entropy (a nonlinear parameter) is the opposite of MV and SD, and it decreases under certain experiment conditions as difficulty level (in the M–L and A–P directions) increases.

Muscle strength is important for stabilizing body position. The results of a previous study showed that an eight-week training program focused on balance helps to adjust and maintain unbalanced strength. This can increase balancing ability [28].

In light of our findings and the opinions of other researchers, we conclude that a heavy load changes postural-stability control. On the other hand, low (and moderate)-load weights do not affect postural control. This discovery is probably due to the fact that the balance system can cope with repetitive daily activities with a low to moderate load (e.g., shopping, wearing and carrying different objects, and changing clothes for different seasons of the year). However, a heavier load causes significant changes in balance control.

Research into heavy loads applied to groups of women of different ages is rare. In light of demographic, social, and economic changes, studies should be carried out on the working-age cohort (taking into account the trend towards a greater proportion of manual workers) and women of retirement age. This is in line with the conclusions of Schiffman et al. [2] and Wojciechowskiej-Maszkowskiej et al. [29] with regard to the need for continuous research into postural control. This research is needed to determine the dynamic characteristics of COP time series in response to applied loads and to minimize the risk of falls.

## 5. Conclusions

The hypothesis of this study was confirmed. A considerable external load of approximately 48% of body weight applied to the bodies of young women (symmetrically distributed) significantly affects postural-stability control. A heavy load results in poorer standing control (increase in SD), increases the dynamics of postural-stability control (change in MV), and requires more attention to control the standing position (change in SE). The absence of a load and smaller loads (22.6%) did not significantly affect body sway.

At the current stage of research, it is difficult to unambiguously assess whether a significant load on the bodies of young women allows them to perform motor tasks without the risk of losing their balance, or whether it causes an interference that may pose a risk of falling. We contend that inferior postural control mainly results from insufficient experience in lifting such a load.

It is also necessary to clarify whether the observed increase in the “instability” of postural control in young women participating in this study precludes any further intervention. For this purpose, the authors plan to carry out further research.

A limitation that needs to be considered when interpreting the results of this study is the small sample size. Possible learning and/or fatiguing effects also cannot be excluded due the absence of a randomized order of testing.

## Figures and Tables

**Figure 1 ijerph-17-00490-f001:**
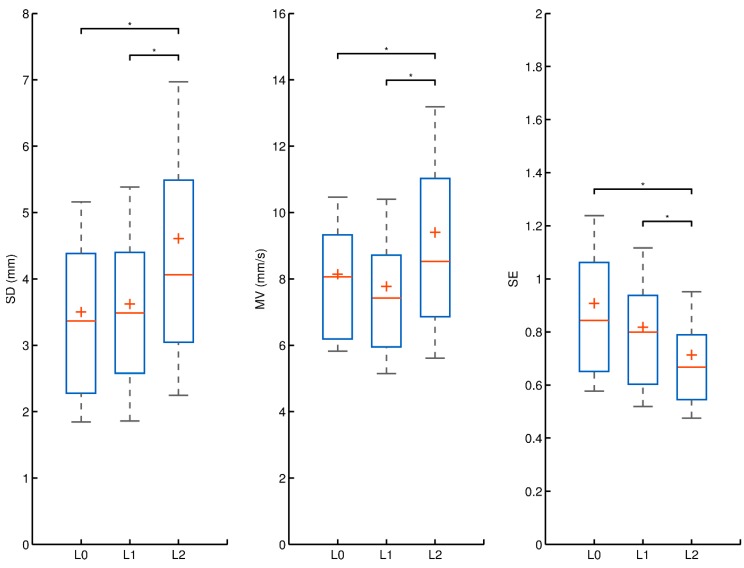
Main effects and significant post hoc tests. Note: +, sample mean; horizontal line, sample median; upper side of rectangle, 75% of data (upper quartile); lower side of rectangle, 25% of data (lower quartile); whiskers, mean ± standard deviation.

**Table 1 ijerph-17-00490-t001:** Differences in center-of-pressure (COP) parameters in individual experiments.

Direction/Parameters	Test		
	L0	L1	L2
M–L			
SD (mm)	2.62 ± 1.13	3.01 ± 1.44	3.65 ± 1.66
MV (mm/s)	6.60 ± 1.63	6.31 ± 2.07	7.69 ± 3.05
SE	0.94 ± 0.37	0.80 ± 0.33	0.71 ± 0.25
A–P			
SD (mm)	4.38 ± 1.65	4.23 ± 1.87	5.56 ± 2.59
MV (mm/s)	9.68 ± 1.84	9.24 ± 2.31	11.11 ± 3.72
SE	0.87 ± 0.28	0.84 ± 0.27	0.72 ± 0.22

Note: M, mean; SD, standard deviation; MV, mean velocity; SE, sample entropy; M–L, medial–lateral; A−P, anterior−posterior; L0, no load; L1, 7 kg load in each hand (14 kg total); L2, 15 kg load in each hand (30 kg total).

**Table 2 ijerph-17-00490-t002:** Main effects and interactions from analysis of variance.

	ANOVA								
Parameters	Load Effect			Direction Effect			Load–Direction Interaction		
	F(2,48)	*p*	η_p_^2^	F(1,24)	*p*	η_p_^2^	F(2,48)	*p*	η_p_^2^
**SD (mm)**	10.33	<0.001 *	0.24	83.98	<0.001 *	0.72	1.65	0.199	0.05
**MV (mm/s)**	9.19	<0.001 *	0.22	310.97	<0.001 *	0.9	1.23	0.3	0.04
**SE**	12.06	<0.001 *	0.27	0.07	0.793	0.01	2.31	0.106	0.07

* Significant difference. Note: F, ratio of variances (dispersions); η_p_^2^, partial eta squared.

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
