# Peer review of "Characteristics of Standing Postural Control in Women under Additional Load"

_ijerph, 2020, doi:10.3390/ijerph17020490_

Round 1
Reviewer 1 Report
Thank you for submitting your effort. I found the subject very interesting and applicable in a few different venues. Your understanding of the topic is obvious. I also appreciate the methods you employed to conduct your study. The discussion is lengthy and provides ample information for the consumer. Please find a few suggestions below that are offered to potentially enhance your product.
Intro -
Be more definitive in your language - change 'seem' to 'is'
The research question section, although informative, does present in a choppy manner. Perhaps rethinking this format for one that flows a bit more.
Methods -
Nicely done!
Results -
Well done also. Perhaps a shift in organization of the information provided under Figure 1. Again, presents turbulently.
Discussion -
Great effort!
Make sure the font is consistent throughout the document. There appears to be a change at a certain point in the document.
References -
Unbold the dates.
Reviewer 2 Report
Dear Authors,
Thank you for this interesting paper.
The paper is in good written, only please, ignore table 1 and write Height & Weight after Age in text.
Be sure the force platform sampling frequency is 100 Hz, it seems too low. The rest of the paper is good. Regards
Reviewer 3 Report
The authors of the submitted article investigated the effect of additional load on posture stability control. 34 young women of a mean age of 20.8 yrs were examined. Postural sway expressed by CoP measurements (variability, mean sway velocity and entropy) obtained from a series of three 30-second tests in quiet stance with no visual restriction, bearing 0 kg, 14 kg and 30 kg additional load. The results of the study showed that a load approximately 48% of body mass led to random movements that eventually resulted in decreased postural control. The authors concluded that the insufficient experience of the participants was a factor that resulted in inferior postural control and suggest the usage of functional training with an equivalent load to improve balance control.
The subject of the study is within the scientific disciplines represented in IJERPH. Nevertheless, some issues have to be addressed in order the manuscript to be accepted, as pointed out in the following General and Specific comments.
General Comments:
Language and syntax should be improved throughout the text. Authors use a variety of non-synonymous terms to describe their experimental variable (load, stress, strain, burden, resistance, charge …). The rationale of the study should be supported by an adequately evidence-based Introduction. For example: Are women more instable of men in the respective occupation as implicated in the first paragraph of the 2nd page? Is this supported by data from the literature? Procedure: A more detailed description is required regarding the body posture, the randomization of testing order and the familiarization of the participants. Loads: A large SD of mean body mass is presented in table 1. The researchers have to justify why the additional load was fixed at 14 kg and 30 kg and why no extra load relative to body mass (i.e. 20% and 50% of the individual body mass of each subject) was implemented. There are factors such as “psycho-physical capabilities” and “muscle strength activity” that are mentioned in the study without any indication that were assessed with the presented methodology. No specific mechanisms that regulate balance and had an effect on the differences observed in the study are discussed. At the Conclusions, a suggestion about functional training is mentioned without any data in the study supporting this notion.
Specific Comments:
Abstract quiet stance using three 30-second tests with participant with having their eyes ⇒ it is suggested to rewrite this sentence. The following weights were used in the study: 14 kg and 30 kg ⇒ actually, there are three loading conditions: 0 kg, 14 kg and 30 kg. we used four linear COP parameters (variability and mean sway velocity and one non-linear parameter: entropy) ⇒ only three parameters are reported within the brackets. requires more attention resources invested to control standing posture ⇒ This mention in the Results section of the Abstract is not supported by the measurements made in the study. Using functional training with an equivalent load could help to improve balance control ⇒ This is speculative, as no such data were acquired in this study can support this statement.Introduction risk of collapse ⇒ risk of fall. Little research has been done ⇒ Limited research. Crommert et al., ⇒ Crommert et al. proprioceptive feeling ⇒ What do you mean by “proprioceptive feeling”? Other researchers ⇒ Cite the references. the effect of additional strain ⇒ Note that, in a biomechanical perspective, strain is the measure of deformation. It is suggested to use the term “load”. young women has not been studied previously ⇒ Check Reference #9 (“There are no significant differences in the response to the amount and configuration of the load between male and female subjects”). Women do harder physical work…people weighing more than 70 kg. ⇒ is there evidence to support this? As this is not a key factor for the design of the study, it is proposed to delete these sentences. the causes of instability ⇒ are women more unstable than men in their working environment? Provide evidence for this statement. can improve planning of physical strain at work ⇒ it is suggested to rewrite this phrase for better understanding of the readers. additional external burden ⇒ additional external load. What influence on the changes in the values of linear and non-linear parameters has a large additional external load on young women? ⇒ it is proposed to rewrite this research question for better clarity. The following hypotheses were proposed ⇒ The following hypotheses were tested. Hypothesis No 1: Combine the two sentences in one.
Materials and Methods the participants were familiarized ⇒ participants were familiarized. A brief description of the familiarization should be included. the participants were young women who achieved maximum psycho-physical capabilities ⇒ How were these capabilities tested? Strength endurance … to working with patients. ⇒ The content of this sentence is not suitable for the Materials and Methods section, but it could add to address the issues mentioned in Comments #13 and #14 (see above). (Nelson et al., 2005) ⇒ It is not mentioned in the Reference list. Also, see journal’s guidelines for the citation of the references in the text. Provide also the inclusion and exclusion criteria for the selection of the participants. Table 1. Characteristics of the subjects (M ± SD) ⇒ Table 1. Characteristics of the participants (Mean ± SD) Force platform (Type 9286AA, Kistler Instrumente AG, Winterthur, Switzerland) sampling frequency: 100 Hz; duration of a single test: 30 sec. ⇒ A force platform (Type 9286AA, Kistler Instrumente AG, Winterthur, Switzerland), operating at a sampling frequency of 100 Hz, was used for data acquisition. The duration of each balance test was 30 sec. The experiment consisted of three trials ⇒ There were three tests (L0, L1, L2). Were there also three trials per set, i.e. L0a, L0b, L0c, etc.? If not, it should be “The experiment consisted of three tests”. Were the tests conducted in a randomized order? How many trials per test? Is just one (1) trial per test adequate? in a free standing position ⇒ at the next paragraph it is stated that “a 14 degree angle between the feet and a distance of 17 cm between the heels”. Thus, this is not a free standing position. Were the participants barefooted? If not, was there any instructions/control of the footwear? that was 22.6% of the mean body weight ⇒ the range of the individual percentage of body mass should be reported due to the large SD reported for the mean body mass. that was 48.4% of the mean body weight ⇒ see the above comment #32. What is the rationale to use the fixed 14 kg and 30 kg extra loads? An explanation for this selection vs. the use of extra loads relative to a specific percentage of body mass in mandatory at this point. The load (weights) was distributed ⇒ The additional load was held. at the trochanters of the larger thigh bones (femurs) ⇒ trochanters of the femurs. What was the grip of the load (weights)? Was the angle of the major upper arm joints (shoulder, elbow, wrist) controlled? their gaze focused on a marker at eye level at a distance of 1.5 m ⇒ What was the shape/brightness/size of the marker? What was the rational of the selection of the distance? one non-linear parameter, namely, entropy ⇒ define the type of entropy as “sample entropy” from its first mention in the text. center of pressure (COP) of the feet calculated ⇒ center of pressure (COP) calculated. in the medial–lateral (M–L) and anterior–posterior (A–P) planes ⇒ in the medial–lateral (M–L) and anterior–posterior (A–P) axis. Based on the COP signal ⇒ Describe the smoothing method used for analyzing the signal. Sample entropy (SE) (a non-linear parameter), however, is a measure ⇒ Sample entropy (SE) however, is a measure. 3 load levels (no load, 14 kg and 30 kg) and 2 COP planes (anterior–posterior and medial–lateral) ⇒ 3 load levels (L0, L1 and L2) and 2 COP axis (anterior–posterior and medial–lateral). the statistical significance level of 0.05 ⇒ the significance level of a = 0.05.
Results Change numbering as “3. Results” instead of “2. Results”. Replace “plane” with “axis”. for the three experimental conditions: L0— free standing; L1—load of 7 kg per hand (total of 14 kg in L1); L2—load of 15 kg per hand (total of 30 kg in L2) ⇒ it is suggested to delete the meaning of the abbreviations as they are defined in the “Materials and Methods” section. This should be also done in the last paragraph of the “Results” section. Table 2: Indicate significant differences due to load and/or axis. Table 2, Notes: Replace Plane with Axis. Table 3. Replace Plane with Axis. Table 3. Report p values with three (3) decimal digits. Figure 1: Mean data are not in agreement with the data presented in Table 1. whiskers represent the mean ± standard deviation ⇒ it is suggested to correct this.
Discussion Replace “plane” with “axis”. Replace “stress” with “load”. The aim of this study was to assess the impact of stress ⇒ The aim of this study was to assess the impact of additional load. the specialist literature ⇒ the respective literature. the body oscillations were the highest values ⇒ the body oscillations were higher burden ⇒ load The results of this study show that young women had better control of their standing position at moderate loads than at higher loads. In this study, the absence of load (freestanding, L0) and small load (L1) did not significantly affect the swaying of the body (SD) ⇒ It is recommended to place this sentence after the first sentence of the paragraph. the absence of load (freestanding, L0) and small load (L1) ⇒ Delete “the absence of load (freestanding, L0)” and start with “Small load (L1)…”. However, the loads, ⇒ However, higher loads, Other researchers concluded that instability increases linearly with increasing external loading (8%, 22% and 54% of body weight) ⇒ Cite the researchers. According to Wegen et al. [21] ⇒ According to van Wegen et al. [21]. The highest load (L2) contributed to an increase in the dynamics of muscle strength activity. ⇒ This is speculative as no force and/or EMG recordings are presented in the study. with little external resistance (L01) ⇒ with little external load (L1). by the effects of muscle fatigue resulting from the application of additional static load ⇒ see comment #66 above. alloys held together ⇒ What do you mean? These discoveries ⇒ These findings. To the best of our knowledge, this study was the first assessment ⇒ Check Reference #9 [Rugelj, D., & Sevšek, F. (2011). The effect of load mass and its placement on postural sway. Applied Ergonomics, 42(6), 860-866].
Conclusions increases the dynamics of posture stability control (change in MV value) ⇒ This is a kinematical rather than a dynamical parameter of postural control. the risk of disrupting and losing their body balance ⇒ the risk of disturbing and losing their body balance.
References Add Nelson et al., 2005.
Round 2
Reviewer 3 Report
In the revised version of the submitted study, the authors provided sufficient answers to the majority of the issues noted in the initial version of the text.
Nevertheless, in the current version, there are points of the initial general comments where no adequate improvements were made, i.e. (numbering refers to the initial review comments):
#1: Language ⇒ although excessively improved, authors should limit the extensive use of text in the brackets that actually contribute to the development of their statements. #3: Evidence based introduction ⇒ although the limited number of research conducted in females, there are interesting findings in the literature concerning the effect of external load on balance parameters that could add to the buildup of the rationale of the study. #6: “maximum psycho-physical capabilities” ⇒ it is still mentioned (L71) without any explanation which are this capabilities, and how participants achieved them. In detail, it has to be clear a) how the mentioned positive medical certificate was acquired by the participants, b) the testing procedure to establish the capability to acquire the certificate, c) which parameters were examined, d) how these parameters are connected to the psycho-physical capabilities, and e) was there a selection procedure, i.e. if a participant had a sub-maximum score at the testing procedure was she excluded from the parameter? #7: As briefly discussed in the Introduction (see also the comments above), it seems that the experimental design is to stimulate the proprioceptive sensory input and to control the visionary input. Yet, a further discussion of the effect of load on proprioception is needed. In detail, readers should get information why the extra additional load differentiates the proprioception input and how this difference is reflected to motor behavior. This could add interesting information to the literature.
In addition, some of the specific comments were not replied, i.e. (numbering again refers to the initial review comments):
#9: proprioceptive feeling ⇒ it is still mentioned (L32). #21: A brief description of the familiarization is not included despite the proposal in the initial review. #27: about the instrumentation (L85-87) ⇒ proper syntax is still needed despite the suggestions made in the initial review. #29: testing procedure ⇒ provide explanation why just one (1) trial per test is adequate. See also the comments on the last paragraph of the review. #41: medial–lateral (M–L) and anterior–posterior (A–P) planes ⇒ despite the references provided by the authors, a plane is a level surface defined by two axis. In the case of posturography, CoP is examined at the transversal plane that is formed by the medial–lateral (M–L) and anterior–posterior It is strongly recommended to adopt this terminology and to provide the axis abbreviations in a constant manner throughout the text (especially in p. 6). #45: the significance level of a = 0.05 ⇒ there is no statistical index reported (L116). #49: Indicate significant differences due to load and/or axis ⇒ not included in Table 1. #72: dynamics of posture stability control ⇒ MV is a kinematical parameter. This is not corrected and mentioned again (L190).
Below are some comments on the current version of the text:
L18: use another word for we suppose (it seems speculative). L40-42: syntax has to be improved. L44: the aim of this study was to evaluate… L52: Research question. L93: height of the trochanters L95: upper extremities instead of hands. L119: M ± SD L140: it is direction of movement, not plane! L147-148: L2 condition is already defined, text in brackets should be deleted. L161-162: “the body oscillations were higher load” ⇒ besides improvement in syntax, what do you mean? L182: study L226: results of a previous study. Add the limitations of the study.
But the main concern for the study is the design and the data analysis. Firstly, with just one trial there is no evidence that a specific motor behavior is solid and not influenced by random incidents. Secondly, possible learning and/or fatiguing effects cannot be out-ruled due the absence of a randomized order of testing, although executing from no load to the highest load is methodologically correct in sports science. There is no evidence provided by the adopted experimental design that “the lack of load and a smaller load did not significantly affect body sway” (L251-252) without any control of i.e. fatiguing and learning effects. Third, using raw data in posturographic measurements without any filtering should be considered along with additional information concerning parameters representing the noise of the recorded signal, the test-retest variability of the measurements etc. These factors are determinant for the evaluation of the submission.
